# Identifying the Gut Virome of *Diaphorina citri* from Florida Groves

**DOI:** 10.3390/insects14020166

**Published:** 2023-02-08

**Authors:** Chun-Yi Lin, Ozgur Batuman, Amit Levy

**Affiliations:** 1Citrus Research and Education Center, University of Florida, Lake Alfred, FL 33850, USA; 2Department of Plant Pathology, University of Florida, Gainesville, FL 32611, USA; 3Southwest Florida Research and Education Center, University of Florida, Immokalee, FL 34142, USA

**Keywords:** Asian citrus psyllid (*Diaphorina citri*; ACP), gut virome, *C*Las, insect-specific viruses, *D. citri-*associated C virus (DcACV), *D. citri* densovirus (DcDV), *D. citri* reovirus (DcRV), *D. citri* flavi-like virus (DcFLV)

## Abstract

**Simple Summary:**

Controlling Asian citrus psyllid (*Diaphorina citri*), the insect vector of *Candidatus* Liberibacter asiaticus (*C*Las), is an important part of Huanglongbing management. Specific viruses were recently discovered in a psyllid host where they may act as biological agents for minimizing psyllid populations or *C*Las transmission. We analyzed the psyllid gut virome from five regions in Florida and identified four *D. citri*-associated viruses at different compositions. The presence of these specific viruses in the gut indicated potential interactions between these viruses and *C*Las. Our results provided evidence that *D. citri*-associated viruses located in the host gut and their potential activities may be useful for pest management.

**Abstract:**

Asian citrus psyllid (*Diaphorina citri*) transmits the bacterial pathogen *Candidatus* Liberibacter asiaticus (*C*Las), the putative causative agent of citrus Huanglongbing disease (HLB). Insect-specific viruses can act against insects as their natural enemies, and recently, several *D. citri*-associated viruses were discovered. The insect gut plays an important role as not only a pool for diverse microbes but also as a physical barrier to prevent the spread of pathogens such as *C*Las. However, there is little evidence of the presence of *D. citri*-associated viruses in the gut and of the interaction between them and *C*Las. Here, we dissected psyllid guts collected from five growing regions in Florida, and the gut virome was analyzed by high throughput sequencing. Four insect viruses, including *D. citri-*associated C virus (DcACV), *D. citri* densovirus (DcDV), *D. citri* reovirus (DcRV), and *D. citri* flavi-like virus (DcFLV), were identified, and their presence in the gut, including an additional *D. citri* cimodo-like virus (DcCLV), were confirmed with PCR-based assays. Microscopic analysis showed that DcFLV infection leads to morphological abnormalities in the nuclear structure in the infected psyllid gut cells. The complex and diverse composition of microbiota in the psyllid gut suggests a possible interaction and dynamics between *C*Las and the *D. citri-*associated viruses. Our study identified various *D. citri*-associated viruses that localized in the psyllid gut and provided more information that helps to evaluate the potential vectors for manipulating *C*Las in the psyllid gut.

## 1. Introduction

Asian citrus psyllid (ACP; *Diaphorina citri* Kuwayama), the insect vector of *Candidatus* Liberibacter asiaticus (*C*Las), has transmitted and spread the citrus Huanglongbing (HLB) disease worldwide [1]. All commercial citrus cultivars are susceptible, and the destructive disease has threatened citrus production in the United States and other countries in the world [2,3]. In the US, the approximately $9 billion citrus industry in Florida, as well as in other states such as Texas, California, and Georgia, is facing challenges due to HLB disease [4,5,6,7,8]. Some treatments of citrus plants, including antibiotic applications, nutritional supplements, heat therapy, and plant-immune inducers, have been shown to somewhat delay the plant decline; however, none of these methods overcomes or cures HLB disease [9,10]. Therefore, control of *D. citri* has become an essential and important issue in HLB disease management. Current control strategies are limited and rely on insecticides to suppress the insect population. Unfortunately, these chemical-based methods are often ineffective because they are non-specific, costly, environmentally unfriendly, and lead to insecticide resistance [11,12,13,14]. Therefore, biological control has been studied and considered to be an effective, specific tool for controlling insect vectors [15,16,17]. The introduction of entomopathogens as a biological control was studied and practiced in fungi, bacteria, viruses, and protozoa [18]. Insect viruses have been used since 1970, to control cotton bollworm [19]. The family *Baculoviridae* is the best-studied and most-used viral agent worldwide for controlling insect pests from the order Lepidoptera, Hymenoptera, and Coleoptera [20,21]. The baculovirus expression vector system (BEVs) was developed for applications such as (1) post-translational modification to introduce gene delivery in mosquito cell types, larvae and adults [22]; and (2) the surface-display platform to surprise foreign proteins on a viral or cell surface [23]. However, as these natural enemies have narrow host ranges, the diversity of novel insect viruses needs to be studied to cover wide spectra of different insect orders.

Insect microbiomes have been studied, especially in the guts. The interaction of insect and its gut-microflora is involved in many biological and physiological alterations such as (a) mating and reproductive control; (b) essential materials and nutrients supply; (c) immunity and protection against pathogens; (d) diet and digestion compensation [24,25,26,27,28]. To find potential candidates, more recently studies of insect virome were conducted [29,30,31,32]. The diversity and abundance of either DNA or RNA viruses in insects were shown via metagenomics analysis. In *D. citri*, at least five novel viruses are known: *Diaphorina citri-*associated C virus (DcACV), *Diaphorina citri* densovirus (DcDV), *Diaphorina citri* flavi-like virus (DcFLV), *Diaphorina citri* picorna-like virus (DcPLV), and *Diaphorina citri* reovirus (DcRV), from the families *Parvoviridae*, *Flaviviridae*, *Iflaviridae*, *Reoviridae*, and an unclassified (+) ssRNA virus [33,34,35,36,37]. In Florida, Britt et al. [15] completed a comprehensive surveys of psyllid populations in the major citrus production regions of 21 counties. The results demonstrated that all these five viruses and another novel reo-like virus (*D. citri* cimodo-like virus, DcCLV) existed in citrus groves [38], and that DcACV was the most prevalent virus (>60%) followed by DcPLV and DcFLV. Additionally, massive reads related to Citrus tristeza virus (CTV) by RNA sequencing were also found in all sample-collecting regions. The results indicated that a high prevalence rate of CTV disease occurred in Florida and the possible acquirement of CTV by *D. citri* [39]. Recent reports supported the possibility that *D. citri* could act as a potential vector of CTV to acquire, persist and transmit [40]. To further understand the characteristics of these *D. citri*-associated viruses in *D. citri*, more studies of these viruses, such as their distribution and how they interact with psyllid gut tissue, are needed.

Studies have illustrated that the midgut acts as the first physical barrier preventing *C*Las movement and invasion of hemolymph [41,42,43]. The observed replication sites of *C*Las in midgut cells highlighted the important role of the midgut [43,44]. Considering that the insect midgut plays a critical role for *C*Las in establishing and colonizing their insect vectors, the involvement of insect-specific virus may affect those pathogens. Rashidi et al. [45] confirmed that both DcACV and DcFLV co-occurred with *C*Las and showed a positive correlation in the psyllid population in Florida groves, and that DcFLV co-localized with *C*Las in the midgut and salivary gland of psyllid. The co-existence of both *C*Las and each *D. citri*-associated virus needs to be confirmed, especially in the midgut tissue, which serves as a reservoir and provides additional biological or molecular functions interacting with both pathogens. The interactions of *C*Las and these viruses and the possible impacts of the interaction on the ability of acquisition, replication, and transmission by psyllid remain unclear.

In this study, the gut virome was studied in psyllids that were collected from five citrus-growing regions of Florida, and novel viral sequences were analyzed via high throughput sequencing. Four *D. citri*-associated viruses (DcACV, DcDV, DcFLV, and DcRV) were annotated from the RNA-seq database, and their presence together with that of DcCLV were confirmed in the midgut tissue of *D. citri* by using PCR and RT-PCR. Using microscopic observation, we found abnormal morphology of nuclei in the DcFLV-infected psyllid gut cell, but not in the DcACV-infected gut cell. The study shows that various *D. citri*-associated viruses exist in the midgut cells of *D. citri*, indicating complex and diverse interactions among these viral pathogens and the host insect. The results provide additional information to determine the potential roles of these insect-specific viruses as biological tools to manage the psyllid population and limit the transmission of HLB disease.

## 2. Materials and Methods

### 2.1. Asian Citrus Psyllid Collection, Gut Dissection and Total RNA Preparation

Adult *D. citri* psyllids were collected from five commercial citrus groves (Winter Garden, WG; Lake Alfred, LA; Lake Wales, LW; Vero Beach, VB; and Immokalee, IM) in four counties (Orange, Polk, Indian River, and Collier) around central and southern Florida. Approximately 200 adult psyllids were randomly sampled from multiple trees in each grove by using an insect vacuum (BioQuip Product, Inc., Compton, CA, USA). Psyllids collected from each region were separated into BugDorm mesh cages that contained pathogen-free citrus plants (*Citrus macrophylla*), at 25 ± 2 °C under a 14 light:10 h dark photoperiod and 60–70% relative humidity in the greenhouse. To obtain whole guts, psyllid adults were put in Eppendorf tubes and sedated on ice for 15 min; each adult was then dissected with sharp forceps in phosphate-buffered saline (1× PBS) solution under a stereomicroscope to isolate the midgut [45]. At least 50 guts were prepared from each region and concentrated in a 2.0 mL Eppendorf tube with 1× PBS solution by centrifugation at 12,000 rpm at room temperature; the solution was then removed, and the guts were stored at −20 °C. Total RNAs for RNA-seq analysis were extracted from 50 dissected gut tissues of adult psyllids from each separate cage, using TRIzol reagent (Thermo Fisher Scientific Inc., Waltham, MA, USA) according to the manufacturer’s instructions. The extracted RNAs were verified for sufficient purity (the value of A260/A280: 1.9–2.0) and quantity (avg. 150 ng/µL) for RNA-seq using a BioDrop Spectrophotometer (Biochrom Ltd., Cambridge, UK).

### 2.2. RNA-Sequencing, Assembly, and Sequence Analyses

RNA samples were sent to the Foundation Plant Service at UC Davis, CA, USA for high-throughput sequencing (HTS) analysis. The HTS analysis was conducted as described by Al Rwahnih et al. [46] and Britt et al. [15]. In brief, aliquots of the total RNA samples were separated and ribosomal RNA depletion and the construction of a cDNA library were performed using TruSeq Stranded Total RNA with RIbo-Zero Plant Kit (Illumina Inc., San Diego, CA, USA). All five samples were sequenced on the Illumina NextSeq 500 platform. To determine the presence of reads from *D. citri* gut-associated viruses and other viruses in the HTS analysis, two analysis methods were used: a) the raw reads were trimmed and were aligned to the viral GenBank database using a PathoScope version 2.0.6 [47]; b) the resulting reads were adapter-trimmed and de novo assembled with a CLC Genomics Workbench (v8.5.1; Qiagen, Hilden, Germany) into contiguous consensus sequences (contigs) with at least 200 bases lengths. Generated contigs were annotated and compared with the viral genome database of the National Center for Biotechnology Information (NCBI) Refseq using tBLASTx (v2.4.0), (http://www.ncbi.nlm.nih.gov/genome/viruses/) accessed on 2 August 2021 [48]. Possible viral sequence candidates were determined while contigs matched to the viral genome database with a combined E-value equal to or less than 10^−4^ [46]. A BLASTn (v2.10.1) tool was further used to align assembled contigs to the GenBank database of the NCBI.

### 2.3. Verification of Viral Sequences in D. citri

A virus-specific reverse transcriptase-polymerase chain reaction (RT-PCR) and quantitative PCR (qPCR) primers were used to validate the presence of these novel viruses in the *D. citri* gut, based on the assembled list of contigs from HTS results. The sequences, annealing temperatures, and product sizes of the detection primers are shown in Table 1. In brief, two-step RT-PCR and the total RNA extracts from *D. citri* gut tissue as described above were used. Total RNA was diluted in 30 µL of RNase- and DNase-free water and used for cDNA synthesis. cDNA was synthesized from 100–200 ng RNA using a MultiScribe™ Reverse Transcriptase Kit (Thermo Fisher Scientific Inc., Waltham, MA, USA) according to the manufacturer’s instructions; the cDNA was then used as a template for novel virus detection by PCR and qPCR, respectively. RT-PCR was performed under these conditions: 95 °C for 2 min, 35 cycles of 95 °C 30 s, 50–60 °C 30 s, 72 °C for 1 min, and a final extension at 72 °C for 5 min. The amplicons were separated in 1% agarose gel with 100 vol for 1 h and visualized in Gel doc (Bio-Techne, Minneapolis, MN, USA) for the presence of novel viruses. qPCR detection of CTV was performed on a 7500 Fast Real-Time PCR system (Applied Biosystems, Waltham, MA, USA) using a Taqman^®^ RNA-to-C_T_^TM^ 1-Step Kit (Applied Biosystems, Waltham, MA, USA) with the CTV-specific primer/probe (Table 1). The reaction was performed in a 12 µL reaction mixture containing 6 µL of 2 × TaqMan^®^ RT-PCR Mix, 0.3 µL of 40× TaqMan^®^ RT Enzyme Mix, 1.25 µL of each primer, 0.1875 µL of the probe and 1 µL of cDNA template. The condition of the qPCR was 50 °C for 15 min, 94 °C for 10 min, 40 cycles at 94 °C for 15 s, and 60 °C for 1 min.

### 2.4. Psyllid Gut Staining with DAPI

Reproductive organs and guts were dissected from healthy, DcACV-infected and DcFLV-infected adults psyllids as mentioned above. After collecting sufficient numbers of dissected guts (20 or more), we washed the guts three times with 1× PBS solution. Using 4“, 6-diamidino-2-phenylindole (DAPI) Fluoromount-G^®^ (Southern Biotechnology Associates, Inc., Birmingham, AL, USA) staining to visualize the nuclei in gut cells. The dissected guts of healthy and DcFLV-infected psyllids were removed from the excess PBS solution, immersed in approximately 50 µL DAPI Fluoromount-G^®^ and were mounted on microscope slides, covered with coverslips. An inverted laboratory microscope Leica DM IL LED (Leica Microsystems Inc., Deerfield, IL, USA) was used for visualization.

## 3. Results

### 3.1. Identification of D. citri-Associated Viruses in the Gut Tissue by HTS and Validation by RT-PCR

The HTS yielded between 13 and 26 million raw reads and generated an average of 41,215 assembled contigs (length > 250 bases) for the five cDNA libraries (Table 2). Assembled sequences were blasted against to the NCBI database and the results showed a similarity to different insect-specific and plant viruses in the gut samples from all five regions (Table 3). Among these, some viral sequences were identical to sequences of five known *D. citri*-associated viruses, including DcACV, DcDV, DcFLV, and DcRV. To validate the presence of these *D. citri*-associated viruses in the psyllid gut tissues from each region, RT-PCR was performed to trace back and compare with the results of the HTS. The results showed a 100% match between RT-PCR and HTS that confirmed the presence of these five associated viruses in our surveyed samples (Table 4). In addition, we detected an additional sequence of DcCLV via RT-PCR in Winter Garden. DcACV was found in all the five regions, while DcRV had the lowest incidence in our survey. Furthermore, both HTS and qPCR analysis showed a high prevalence of CTV sequences in the Florida regions as well as *Candidatus* Liberibacter asiaticus (*C*Las) (Table 4). More than 40% of the total reads were matched to the CTV genome in three regions; however, less than 3% matches were observed in the other two regions (Figure 1).

### 3.2. Nuclear Damage of Psyllid Adult Guts under D. citri-Associated Virus Infection

DcACV- and DcFLV- infected insects were observed for the changes in their gut and nuclei morphologies. Compared to the guts from healthy (Figure 2A) psyllids, we observed dark and necrotic-like spots in DcACV- (Figure 2B) and DcFLV-infected psyllid guts (Figure 2C). To investigate the impacts of viral infection on gut cell morphology and nuclear structures, DAPI staining was used on healthy, DcACV- and DcFLV-infected psyllid guts. The stained guts from DcFLV-infected psyllids showed irregular shapes and abnormal structures (Figure 2F), indicating the occurrence of pyknosis and karyorrhexis. In contrast, nuclei from healthy and DcACV-infected psyllids appeared in uniform shapes and normal structures (Figure 2D,E). The morphological abnormalities of nuclei in the gut cells were observed in 41.2% of DcFLV-infected psyllids (total 913 nuclei), while low percentages of nuclei abnormalities were observed in both healthy (18.2%; total 793 nuclei) and DcACV-infected (20.8%; total 378 nuclei) psyllids.

## 4. Discussion

Biological control has been considered as an effective, persistent, eco-friendly option for pathogen/pest control. In particular, in the field of pest control, potential candidates such as bacteria, fungi, viruses, or endosymbionts were evaluated for their ability and suitability as biological agents [16,20,50,51,52]. Using insect-specific viruses (ISVs) as biological control agents has become a valuable method to control vector-borne diseases [21,36,53,54,55,56]. However, due to their narrow host range, low pathogenicity [57], and the lack of proper cell lines, it is difficult to study insect pests and their viruses. How *D. citri*-specific pathogens affect their host and, more importantly, how they interfere with *C*Las, remain unknown. Thus, determining the suitable *D. citri*-associated virus that can serve as expression vectors can be a first step to providing us with a useful tool to deliver desirable traits into *D. citri* for further study and control.

Our previous studies showed that DcFLV is systemically distributed throughout *D. citri* and can be vertically transmitted to offspring. The association between DcFLV and *C*Las was determined by showing that their incidences are positively correlated and both pathogens co-localize in the midgut [45]. These results suggest that these novel *D. citri*-associated viruses may not only infect the insect host, but also interact with *C*Las in the midgut. In this study, HTS results confirmed that four viruses exist in the midgut and indicated that the virus composition in Florida varies in different regions. This composition seems to be much more complex and diverse based on many factors such as climate condition, plant cultivars, and the ecosystem. It also further helps us to identify and determine the biological aspects and possible interaction with the host because the midgut serves as a critical barrier for the replication and transmission of insect-borne pathogens [42,58,59]. It is intriguing to understand how these psyllid-specific viruses, either DNA- or RNA-based, face host defense machinery and interact with other gut symbionts or pathogens such as *C*Las or CTV. We studied the gut pathogenicity of DcFLV and DcACV, in a single infection in each surveyed region, on the insect host. The microscopic results showed that DcFLV infection can cause gut damage, particularly the nuclear morphology in cells. The necrotic-like symptoms were similar to the process of apoptosis, including nuclear pyknosis and karyorrhexis. In the DcFLV-infected guts, moderate percentages (41.2%) of the nuclear disruption occurred, compared to the normal cell cycle in healthy (18.2%) or DcACV-infected guts (20.8%). The results indicated that DcFLV may have mild virulence to its native insect host. Ghanim et al. [42] previously showed that similar symptoms were observed in the *C*Las-infected psyllid midgut with a higher occurrence (~65%), but not in the midguts of the potato psyllid-infecting *Candidatus* Liberibacter solanacerarun (*C*Lso) [60]. The results suggest that either the bacterium or the virus induces different and adaptive levels of immune responses in its insect hosts, leading to cell apoptosis to limit the transmission of pathogens. Although the results indicated the potential pathogenicity of both *D. citri*-associated virus and *C*Las to the psyllid gut tissue, we still need more evidence to clarify if either synergism or attenuation occurs when both pathogens exist at the same time. It is critical to understand whether the ISV plays a simple role as a biological agent, or alternatively helps *C*Las colonization and spread. Based on our HTS and PCR results, it is clear that mix infections of *D. citri*-associated viruses commonly exist in psyllid populations. It will be interesting to further measure whether additive or competitive effects have an impact on psyllids infected with multiple pathogens.

Insect viral-based expression vectors have been deployed in a wide range of applications; for example, the baculovirus expression vectors system was developed to apply complex molecule production, potential vaccines, and other therapeutic proteins [23]. Because of their being well adapted to insect hosts, ISVs may interfere with the vector transmission of the arbovirus pathogen, the causative agents among humans and animals [61]. ISVs can also be useful to induce different traits in their hosts [37], such as using virus-induced gene silencing (VIGS) to target specific insect RNAs and using engineered viruses to induce host immune response against other pathogens [62,63]. In addition, recombinant ISVs, such as *Flaviviridae*, can be vector platforms for vaccine development due to their ability to induce immune responses, high stability, and large-scale manufacturing [64,65]. From our studies it is clear that DcFLV can evenly distribute inside psyllids, causing damage to the midgut cells, and shows association with *C*Las, making it a potential candidate as a biological agent. However, its large genome (~27,000 bp) can limit its usage to perform as an expression vector. DcACV and DcDV may be potential candidates for further investigation, because of a) the most common existence (100%) of DcACV in the field, indicating that it is highly contagious in psyllid populations and b) DcDV’s relatively small DNA genome (~6000 bp) may be easily engineered as an expression platform.

In our HTS analysis, several insect and plant viruses were also assembled and annotated. As in the previous survey [15,38], the Citrus tristeza virus (CTV) seems to commonly exist inside psyllids in four regions explored in this study, but was absent from the grove in Vero Beach, Indian River County. The phenomena suggest that psyllids may easily uptake CTV via phloem-feeding [15], indicating that *D. citri* also acts as a potential vector of CTV [40]. Based on these findings, the possible interaction between CTV and the psyllid-associated viruses will be an intriguing topic and more studies need to be conducted. Moreover, in the HTS results on the gut virome, we also detected virus-like sequences, known to be associated with Culex mosquito (Culex mononega-like virus 2), earwig (Hubei earwig virus 1), and cabbage looper (Trichoplusia ni TED virus), but no viruses that are associated with insects from the order Hemiptera (Table 3).

In conclusion, we collected *D. citri* adults from five counties in Florida and dissected the midguts for HTS analysis. The results of *D. citri* gut virome showed the presence of partial sequences of four known *D. citri*-associated viruses, and their presence in the guts was further verified by PCR or RT-PCR. The pathogenicity of two viruses was tested and only DcFLV has the ability to harm the insect gut tissue, resulting in an abnormal morphology of the nuclear in shape and structure. The gut organ, which acts as a diversity pool containing various microbiota, plays a critical role and platform for us to investigate the impact of invaders and the inevitable interaction between different pathogens. The study demonstrates initial and basic groundworks of *D. citri*-specific viruses in the gut organ and provides additional information on the biological roles of these viruses and the potential of biological agents to control the psyllid populations.

## Figures and Tables

**Figure 1 insects-14-00166-f001:**
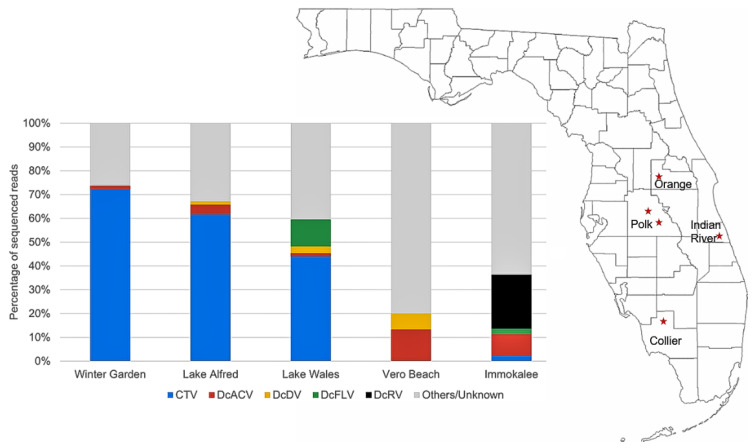
The percentage of viral sequence reads found in the HTS analysis of psyllid guts from each citrus production region in Florida. The composite sample represents the sequences that matched to known insect or plant virus species or unknown viruses.

**Figure 2 insects-14-00166-f002:**
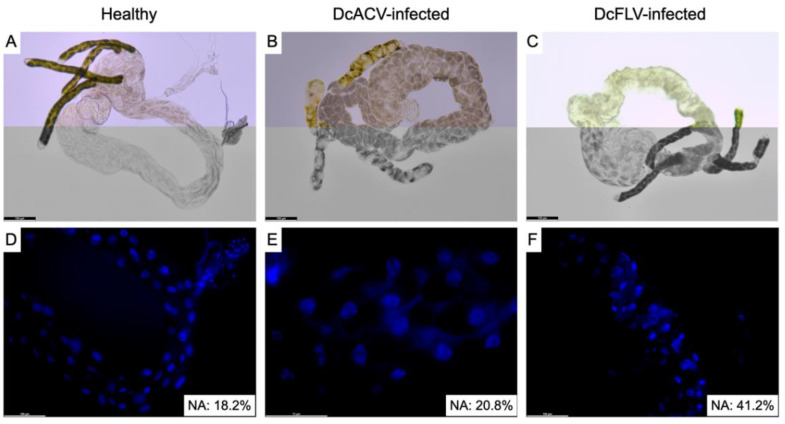
Microscopic analysis of morphological abnormalities of nuclei in gut cells dissected from healthy, DcACV- and DcFLV-infected *D. citri* adults. (**A**) Light micrograph of dissected healthy gut showed normal morphology under color and black/white modes. Light micrographs of the dissected guts from DcACV- (**B**) and DcFLV-infected (**C**) guts showed dark or brown portions. The magnification photographs of dissected guts from healthy (**D**) and DcACV-infected (**E**) showed normal nuclei shapes and structures. The magnification image of the dissected gut from DcFLV-infected gut (**F**) showed that cell nuclei were irregular in shape and vacant in structure. Percentage of nuclei abnormalities (NA) of each three dissected guts (healthy, DcACV-, DcFLV-infected) is labeled at the bottom.

**Table 1 insects-14-00166-t001:** List of detection primers and probes specific to *D. citri*-associated virus, and CTV used in the detection and quantification assays.

Virus ^1^	Primers	Length of Amplicon	Tm (°C)	Accession Number	Reference
DcACV	F-5′ GCCGCACGAAACTAGTGATAAACGCA 3′	473 bp	50	KX235518.1	[15]
	R-5′ GGATCGGTGGTGCACGAGTATGTAAGTA 3′				
DcCLV	F-5′ ATTTAGGGCCATGTGCAAAG 3′ R-3′ CCAACACACCGAGCATACAC 3′	526 bp	62	MZ484733.1	
DcDV	F-5′ AGTCGGTGAGACTGATATCTTCGAGACC 3′	1068 bp	60	KX165268	
	R-5′ GTTTAGTTCGCTTGTCGGTTACACAGG 3′				
DcFLV	F-5′ AGGCGAGTACTCCCATCGGATACATT 3′	1391 bp	58	KX267823.1	
	R-5′ GAGGGCCGCTAAGTCTGTAGGACATATT 3′				
DcPLV	F-5′ TAGGTGAACGTGATAATCCTGGTAT 3′	698 bp	62	KT698837.1	
	R-5′ CAGAACGTCTGTTATGAATCGGAC 3′				
DcRV	F-5′ TTTTCCCAGGTACATCGA 3′	900 bp	50	KT698831.1	[36]
	R-5′ ACCATTCAGCCAGTCCTA 3′				
CTV	F-5′ ACCGGAGCTGGCTTGACTGAT 3′	113 bp	60	MZ670758.1	[49]
	R-5′ CCAAGCTGCCTGACATTAGTAA 3′				
	Probe: 6-Fam/AGAGTGTGCTGTGTACATACAAGCTAAAGA				

^1^ DcACV, *Diaphorina citri-*associated C virus; DcCLV, *Diaphorina citri* Cimodo-like virus; DcDV, *Diaphorina citri* densovirus; DcFLV, *Diaphorina citri* flavi-like virus; DcPLV, *Diaphorina citri* Picorna-like virus; DcRV, *Diaphorina citri* reovirus; CTV, Citrus tristeza virus.

**Table 2 insects-14-00166-t002:** Statistics for *D. citri* RNA-seq assemblies from five regions in Florida.

County	Region	No. of Read Sequences	Megabases	No. of Contigs	BLASTN Virus Contigs	Known Viruses Detected
Orange	Winter Garden	16,748,063	1256	46,344	115	17
Collier	Immokalee	26,004,558	1950	22,670	21	12
Polk	Lake Alfred	20,340,968	1526	49,651	58	19
Indian River	Vero Beach	18,793,456	1410	54,646	8	14
Polk	Lake Wales	12,836,297	963	32,765	99	17

**Table 3 insects-14-00166-t003:** Insect specific virus and Citrus tristeza virus sequences from *D. citri* gut RNA-seq in five Florida regions identified by alignment against NCBI database.

Virus Name	Region ^1^	Reference Accession Number	Longest Contig Length (bp)	Identity (%)	Coverage (%)
*Diaphorina citri* associated C virus	WG, LA, LW, VB, IK	KX235518.1-19.1	2376	98.5–99.9	99–100
*Diaphorina citri* densovirus	LA, LW, VB	YP_009256210.1-11.1	679	34.6–85.5	21–88
*Diaphorina citri* flavi-like virus	LW, IK	KX267823.1	27709	95.2–100	99–100
*Diaphorina citri* reovirus	IK	KT698830.1-36.1	4266	94.3–98.8	88–100
Shuangao insect virus 7	WG, LA, VB	YP_009179392.1	136	33.0	26
Wuhan insect virus 19	WG, LW	YP_009342322.1	4134	37.1–41.8	74–98
Culex mononega-like virus ^2^	WG, LW	ASA47292.1	2079	44.8–46.7	43–81
Photinus pyralis orthomyxo-like virus ^2^	LA, LW, VB	AVR52573.1	710	25.1–32.5	32–91
Liberibacter phage SGCA5-1	LA	KX879601.1	36022	99.5	100
Wolbachia phage WO	LA	KX522565.1	65653	90.5	100
Trichoplusia ni TED virus	VB	YP_009507248.1	1084	46.7	95
Hubei earwig virus ^1^	LW	APG77904.1	758	52.4	58
Lampyris noctiluca errantivirus ^1^	LW	QBP37036.1	1096	49.6	87
Citrus tristeza virus	WG, LA, LW, VB, IK	MK018120.1^2^	19293	89.1–100	89–100

^1^ WG, Winter Garden; LA, Lake Alfred; LW, Lake Wales; VB, Vero Beach; IK, Immokalee. ^2^ Reference accession number: MK779711.1; KC517492.1; FJ525434.1; MF595989.1; KR263170.1; EU937520.1; MH279618.1; KU589213.1; HQ662670.1; GQ338588.1; KP268361.1; MN580434.1; AF260651.1; Y18420.1; JX266713.1; FM955925.1; AY65290.1; DQ272579.1; HM573451.

**Table 4 insects-14-00166-t004:** The Florida regional presence of *D. citri*−associated viruses, Citrus tristeza virus (CTV) and *Candidatus* Liberibacter asiaticus (*C*Las) in composite psyllid gut samples.

Region	DcACV	DcCLV	DcDV	DcFLV	DcPLV	DcRV	CTV	*C*Las
Winter Garden	V ^1^/+ ^2^	X/+	X/−	X/−	X/−	X/−	V/+	+
Lake Alfred	V/+	X/−	X/−	X/−	X/−	X/−	V/+	+
Lake Wales	V/+	X/−	V/+	V/+	X/−	X/−	V/+	+
Vero Beach	V/+	X/−	V/+	X/−	X/−	X/−	X/−	+
Immokalee	V/+	X/−	X/−	V/+	X/−	V/+	V/+	+
Avg. Percentage	100%	20%	40%	40%	0%	20%	80%	100%

^1^ V: Viral contigs were detected and annotated with BLASTn; X: No viral contigs were detected. ^2^ +: Signals were detected by RT−PCR or qRT−PCR; −: No signals were detected. Abbreviations: DcACV: *Diaphorina citri* associated C virus; DcCLV: *Diaphorina citri* Cimodo−like virus; DcDV: *Diaphorina citri* densovirus; DcFLV: *Diaphorina citri* flavi−like virus; DcPLV: *Diaphorina citri* Picorna−like virus; DcRV: *Diaphorina citri* reovirus; CTV: Citrus tristeza virus; *C*Las: *Candidatus* Liberibacter asiaticus. WG, Winter Garden; LA, Lake Alfred; LW, Lake Wales; VB, Vero Beach; IK, Immokalee.

## Data Availability

Data will be shared upon request to the corresponding authors.

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
