# Peer review of "Identifying the Gut Virome of Diaphorina citri from Florida Groves"

_insects, 2023, doi:10.3390/insects14020166_

Round 1

Reviewer 1 Report (Previous Reviewer 1)

The manuscript "Identifying the gut virome of Diaphorina citri from Florida groves" is overall a nice addition to the body of knowledge regarding the diseases associated with the pest insect ACP.

Overall, I believe it merits publication, but the clarity of writing needs to be considerably improved. I offer my main suggestions below, which hopefully can help authors prepare a more finalized version for publication.

Mainly, I believe reference [66] is central to the context of this paper, and should have been presented already in the introduction, instead of just at the very end of discussion. That, and another cited study from Florida [15], had already reported on the abundance of CTV among sequenced samples of ACP, therefore the same observation here was not factually so 'unexpected' as stated in line 84. At this point, I suggest the authors clearly describe how their results build on these previous observations, from start.

The narrative could be significantly condensed by such approach, and I feel its readership would benefit of further considerable revision. For instance, several passages seem confusing, needing reformulation. I list some examples below:

- Lines 60-64;

- Lines 307-310; 

- Lines 312-314;

- Lines 343-345;

 - Lines 363-365.

- The use of 'etc' at line 370 felt innapropriately vague, please make the intended point more clearly.

Finally, perhaps due to generated .pdf resolution for review, all images in Figures are currently hard to interpret in details. Particularly the microscopy images are central to interpreting these observations, and should be thus considerably improved for review.

Thanks for the opportunity, and  looking forwards to seeing a revised version of this manuscript.

Author Response

Reviewer 2 Report (Previous Reviewer 3)

Dear authors,

I was pleased to see the adaptations in your manuscript following the reviewers' comments. Although the goal of the research is now clearly stated in the introduction and discussion, I had to observe that the contribution of the experimental results to the research area is limited. Therefore, I am not convinced by the results that should either show an interaction between insect-specific viruses found in the gut or with CLas. Furthermore, I could not find convincing data to support the potential roles of these ISV as biological tools to manage the psyllid population which was stated as additional information provided by this study.

Author Response

Point 1: I was pleased to see the adaptations in your manuscript following the reviewers' comments. Although the goal of the research is now clearly stated in the introduction and discussion, I had to observe that the contribution of the experimental results to the research area is limited. Therefore, I am not convinced by the results that should either show an interaction between insect-specific viruses found in the gut or with CLas. Furthermore, I could not find convincing data to support the potential roles of these ISV as biological tools to manage the psyllid population which was stated as additional information provided by this study.    Response 1: Thank you. The major goal of the study is to understand the gut virome of D. citri and its associated viruses in different groves in Florida. This is a critical step in the process of developing ISV as biological tools. Since CLas resides in the gut, and the gut is the first barrier for CLas infection of the psyllid, it was important to study the gut associated viruses. Based on the virome analysis, the following objective is to evaluate potential viruses as biological tool to control psyllid. Therefore, we are suggesting DcACV (100% existing in psyllid population) and DcFLV (biology study referenced by Rashidi et al. 2022), for further research. Our previous study (Rashidi et al. 2022) provides statistical result that these two viruses have positive cooccurances related to CLas in Florida groves, indicating the possible interaction exists between D. citri-associated virus and CLas. Furthermore, in this study, we added microscopic results showing that DcFLV could damage nuclear structure in gut cell, whereas no significant cell damage was found in DcACV-infected gut (Figure 2). We think these result are the first evidence to indicate DcFLV has the potential ability to control the psyllid population. More experiments such as behavior, morphology and CLas interaction are in progress to provide evidence to evaluate the biological role of the virus, but are beyond the scope of this paper.        

Reviewer 3 Report (Previous Reviewer 4)

The manuscript has been much improved, and issues raised in comments on previous version were basically addressed. Still, authors are encouraged to further improve the manuscript:

1) Fgure 1 should be replaced with a higher resolution drawings.

2) Use of virus name: according to the lastest version of virus nomenclature, scientific name of a virus species should not be in italics. So  in Line 84 and last line of Table 3 (Line 261-262), name of Citrus tristeza virus should not be in italics. Please check other virus scientific names for such use.

3) Still, there are some typos or errors in the manuscript, such as Line 219, "D. citri-associated virus(es) infection"

Author Response

This manuscript is a resubmission of an earlier submission. The following is a list of the peer review reports and author responses from that submission.

Round 1

Reviewer 1 Report

Comments on the manuscript "Identifying the gut virome of Diaphorina citri from Florida groves" by Lin et al. 

The manuscript reports on the incidence of various viruses in the midguts of Asian citrus psyllids in the state of Florida, US. As part of their finds, the authors highlight the high prevalence of the important citrus pathogen CTV detected from their samples, which is described to be vectored by another distant insect species. The authors claim their finds can be important for developing biocontrol strategies, pending further characterization of the nature of these host-virus relationships.

Overall, the manuscript follows current methods, and reads clearly presented. However, it seems limited in significance in the direction proposed, as it stops at reporting on the diversity of viruses which are potentially associated with the psyllid.

The microscopy images obtained do not clearly confirm the physical presence of some detected virus, and was presented as suggestive of even further uncharacterized diversity. It is hard for a non-specialist to state, but perhaps the interpretation of these features as viral fragments might have to be confirmed. For instance. the first image shows granules measuring several several micrometers; what could that be, exactly?

The emphasis on the prevalence of CTV is an interesting observation. The authors are certainly aware of another publication which has reported on the same phenomenon elsewhere, by Wu et al. (2021). That report is directly associated with the statement on lines 262-264, which claims "the phenomena suggest psyllids can easily acquire CTV during phloem-feeding and more studies are needed to understand (...)", therefore it ought to be mentioned and commented on. Nonetheless, it is not clear how CTV or any of the reported viruses can be used as biocontrol agents, and this is a fundamental point warranting clarification.

Therefore, I do not think the results of the manuscript bring "light on the possible relationship with the host", nor seem to provide "additional information on the biological roles of these viruses and the potential candidates of biological agents for controlling the psyllid populations". As a result, I cannot recommend the present manuscript in its present state. My recommendation is that the authors include more pertinent citations of related reports and clarify how the findings advance the perspectives of biological control of psyllids.  

Hopefully, these comments may help authors improve the manuscript.

Reference:

Wu et al. 2021 https://doi.org/10.3390%2Finsects12080735

Reviewer 2 Report

The paper adds information on the distribution of viruses associated with  Diaphorina citri in Florida.  What will be  even more important is to conduct research on the impact of these viruses on the phenology of infected psyllids. 

Reviewer 3 Report

Dear Authors,

Your manuscript is well written and presents the virome analysis of the gut of Diaphorina citri from five regions in Florida groves. The analysis presents a HTS of RNA from gut tissue of D. citri for virus identification including validation by RT-PCR. Furthermore, HTS complemented with qPCR analysis showed the prevalence of CTV, the Citrus tristeza virus. In addition, viral like structures were observed in the gut cells of D. citri but no further research was undertaken.

While I do think the paper is very well written and is based on a decent analysis with no clearly identifiable flaws; I do have my doubt about the scope/contributions of this research paper. Even though unravelling the gut virome of D. citri have not been carried out before, to the best of my knowledge, the detection and surveillance of D. citri associated viruses in Florida citrus grove populations have been investigated (Britt et al. 2020) and therefore the additional value of this research are very limited in my view.

If you are able to more clearly spell out the more distinguishing and important implications of their virome analysis of the gut of D. citri (e.g. how the viral load behave throughout life stages, in different tissues, which viruses were identified by TEM, which viruses from this analysis can be used for pest control), you could have a solid paper with more additional scientific value.

Sincerely,

Reviewer 4 Report

This paper reports identifying results of 4 insect viruses from guts of Asian citrus psyllid (Diaphorina citri) collected from 5 regions of Florida, USA, first by high throughput sequencing, and then confirmed with RT-PCR. The experimental design was reasonable, and the results are basically interesting to hopefully help understand interactions of psyllids, causal agent bacterium of Huanglongbing, and gut viruses, with potential of D. citri-associated viruses for use of biological control. Here I have two concerns:

1)  4 viruses were identified in psyllid guts in this study,  but no any biological evaluations of these viruses were conducted. So it is too early to claim that these viruses can be of potential use for biological control. 

2) I suggest authors to remove TEM result in 3.2 because this result had very weak or uncertain link to 4 viruses found in the gut. 

In addition, authors are advised to check the thorough manuscript for writing (e.g., Line 81, to establish and colonizing insect vectors)